# A Model to Prevent Substance Use/Abuse by Student Nurses at Limpopo College of Nursing, South Africa

**DOI:** 10.3390/healthcare11162285

**Published:** 2023-08-14

**Authors:** Matodzi Doris Netshiswinzhe, Dorah Ursula Ramathuba, Tsakani Rachel Lebese, Lufuno Makhado

**Affiliations:** 1Advanced Nursing Science, Faculty of Health Sciences, University of Venda, Thohoyandou 0950, Limpopo, South Africa; mdnetshiswinzhe@gmail.com; 2Research Office, Faculty of Health Sciences, University of Venda, Thohoyandou 0950, Limpopo, South Africa; rachel.lebese@univen.ac.za; 3Public Health, School of Health Sciences, University of Venda, Thohoyandou 0950, Limpopo, South Africa; lufuno.makhado@univen.ac.za

**Keywords:** model, prevention, substance abuse, student nurses, nursing college

## Abstract

Background: Substance use/abuse is a global challenge that has detrimental effects on nations’ health, wealth, and security. Substance users in Africa make up roughly 17% to 21% of global illicit drug users in Africa and cannabis abuse (63%) (UNODC, 2010). Purpose: We aimed to develop, describe, and evaluate a model that could be used as a framework of reference to prevent substance use/abuse by student nurses at Limpopo College of Nursing, South Africa. Method: A mixed-method study approach was used following a qualitative exploratory, descriptive, and contextual design to explore factors contributing to substance use/abuse by student nurses and quantitatively examine the impact of substance use/abuse on students’ academic achievements in Limpopo Province. Findings: The results of the first empirical phase reveal two themes and seven subthemes from the semi-structured interviews with lecturers, students, and support staff. The literature supported the results. In phase 2, we analyzed the concept of the “prevention” of substance use/abuse by student nurses following the process of a concept analysis by Walker and Avant (2016). The results were conceptualized within the six elements of practice theory: context, agent, recipient, dynamic, process and procedure, and outcome. Conclusions: The relational statements provided the basis for the model description. A reliable method was used to describe and evaluate the model.

## 1. Introduction

Substance use/abuse among students in tertiary institutions is a social problem that is escalating at a tremendous rate of 62.7%, and most studies have highlighted it as a normal part of the college or university experience; most students abuse substances in the form of smoking cannabis (46%), ecstasy (5.3%), and drinking alcohol (80.6%). Mbuthia et al. [1] indicate that alcohol and substance abuse among undergraduate students is a reality, and alcohol is the most commonly abused substance. Other abused substances include marijuana and cigarettes. Furthermore, Gupta et al. [2], in their study conducted in North India among male college students, revealed that substance use was highest (76.2%) among law students, followed by art students (62.5%), and (53.5%) were alcohol users, followed by smokers (27.3%) and tobacco chewers (8.2%).

If not prevented effectively, such substances may negatively affect the quality of life of students still in the developmental stage. Student nurses come to the college with different social, economic, and cultural backgrounds and academic status. Separated from their families and friends, they find the college environment different from the places where they were raised. In the academic setting, they face new challenges they have not been exposed to before. For instance, sharing single rooms, competition among themselves, and class schedules may lead to stress in nursing students [3,4,5].

In such situations, some students may adopt new coping tactics, such as substance use/abuse, which may have serious consequences [3,6,7]. The rate of risky drinking episodes has been found to be high during the first year of college life as students are trying to cope with the new environment. Student nurses are vulnerable to substance abuse due to the ease of access to outlets such as taverns and beer halls [8,9]. Some student nurse residences are not too far from liquor outlets, indicating easy access to alcohol and other substances.

Nurse career training is recognized as a stressor, often encompassing a traumatic experience during the four years of training; hence, substance use is resorted to as a coping mechanism. Peer pressure, low self-esteem, traditional practices and values, role modeling and imitation, and financial independence influence substance use/abuse [10,11,12,13,14]. The most commonly used substances are alcohol, cigarettes, marijuana, hubbly bubbly/hookah/water pipe (an oriental tobacco pipe which is flavored), and other stimulants. Repeated use of substances can impair decision making and affect judgment. This could be disastrous as student nurses are health professionals who deal directly with human lives, and substance use in this population may lead to medico-legal hazards and lawsuits [3,4,5,15]. Substance use/abuse may significantly impact students’ academic/work performance and ability to cope with their workload [16,17].

The consequences of substance use/abuse include injuries, destruction of property, moral decadence, misallocation of resources, indiscipline, compromised academic standards, termination, and loss of lives [10,18,19,20,21,22]. Students’ substance abuse problems can manifest through absenteeism, tardiness, lower grades in theory, and incompetence in clinical facilities.

Strategies to curb substance use/abuse require collective action; family, college, and the community may be effective preventive measures. The developed model requires a collaborative effort and effective coordination among various stakeholders. This model calls for all concerned stakeholders to adopt a multidisciplinary approach. Each stakeholder must agree on their preventive effort role [23,24,25].

## 2. Setting

Limpopo is a poor rural province with one central, provincial nursing college and five satellite campuses in districts with inadequate health care facilities. The health services where students are placed are marked by shortages of human resources, poor infrastructure, and a lack of resourced medical equipment, which negatively impact the teaching and learning of students, as they sometimes have to bear the brunt of staff shortages.

The status quo of colleges in rural provinces is not fully established in that they lack educational support facilities as well as social or recreational facilities where students can be engaged in their free time or space, unlike colleges in urban settings where there are library facilities and internet connectivity for study purpose and students can de-stress with recreational activities such as playing mentally and physically engaging games. There are no televisions at the student residence, forcing students to seek recreation outside the campus and engage in the use and abuse of substances.

### 2.1. Research Problem

Substance use/abuse by students, and subsequently their addiction to these substances, is a social problem and a public health issue at Limpopo College of Nursing (LCN). A survey by the Department of Health (1998) in South Africa found that approximately 8.3 million South Africans aged 15 years or older consume alcohol regularly. In another investigation on a sample of South African students, it was reported that 80% of participants drank large quantities of alcohol [26].

The researcher observed a trend of absenteeism, with almost 5% reported every quarter during student placement at clinical health facilities, and deficient performance by student nurses. An inquiry was made, and it was noted with concern that the students were abusing/using alcohol, which was reflected in students’ progress reports. The house wardens reported substance use incidences, indicating that students sneak around with alcohol and are always involved in physical brawls with safety officers around the college campus. This is not an isolated problem; the prevalence of substance use is of great concern. A study conducted among university students in South Africa revealed a substance use prevalence rate of 62.7%. The most prominent substances used by students were alcohol (80.6%), cannabis (46%) and ecstasy (5.3%) [1]. Another study conducted among high school students in Limpopo indicated that 35.5% of males and 29.7% of females used alcohol and consumed six or more alcohol units (binge drinking) within 30 days; on one occasion, the consumption was 17.5% and 15.9%, respectively [27,28].

The prevalence of substance use/abuse by the student nurses in the LCN inflicts a social, health, and economic burden on them, including minimizing their full academic potential. Due to poor academic performance, most students need more than the required number of years to complete their studies. Chauke et al. [29] also concur that alcohol use negatively influences schoolwork (e.g., through absenteeism, low performance, truancy, and delinquency). In contrast, others become pregnant and become parents without intention before completing their studies [3,14].

### 2.2. Research Purpose

This research study aimed to develop, describe, and evaluate a model that can be used as a framework of reference for the lecturers and clinical facilitators at nursing education institutions to support nursing students in coping with their academic program and developing into independent, moral, and competent practitioners.

### 2.3. Research Method

A convergent mixed method which was exploratory, descriptive, and contextual in design was utilized to develop the model [29,30,31,32]. The findings from the convergence parallel design reveal that socialization and environmental factors were related to substance use/abuse among student nurses, earning stipends, and lack of necessary support. The findings of the present study reveal that the mean age of students abusing substances is 23 years, which suggests that it is not a recently developed behavior, and the students are not first-time users. Prevention was identified as the key to reducing substance use/abuse among student nurses at LCN. This was conceptualized considering [33] in phase 2, and the concepts were classified in phase 3 of model development by [34], where the model was described, and lastly, the model was evaluated using the method from [30].

### 2.4. Ethical Considerations

Ethical clearance was granted by the University of Venda Ethics Committee (SHS/17/PDC/11/0505). The Provincial Department of Health and Vice Principals of all five campuses granted permission. Participants’ rights were respected in that they were provided with insightful information about the research purpose before signing their informed consent. Afterward, they signed the consent form, agreeing to participate voluntarily. Furthermore, they were assured their right to privacy, anonymity, and confidentiality would be protected throughout the research study.

### 2.5. Development of the Model

Model development was based on Dickoff et al.’s (1968) theoretical framework, where the results of concept analysis were integrated within the theory. Table 1 depicts the findings for the empirical phase [5], which reported the prevalence of substance use/abuse by student nurses at Limpopo College of Nursing.

Structural Description of the Model

The structure of the model depicts the relationships between the agents, recipients, and the environment, also outlining the dynamics and processes.

Model Outcome

The aim is to provide an intervention model to prevent substance use/abuse by students in tertiary institutions.

#### 2.5.1. Assumptions of the Model 

The researchers made assumptions regarding the prevention of substance use/abuse by the student nurses at LCN based on the following statements:-The Department of Health at the LCN can contribute significantly to the prevention of substance use/abuse through a policy framework.-Family plays a substantial role in the primary prevention of substance use/abuse by student nurses in LCN.

The model also assumes that preventing substance use/abuse by students at LCN can result in responsible future professional nurses, with fewer medico-legal hazards and increased quality of patient care overall.

#### 2.5.2. Relational Statements

Relational statements provide a framework for guiding the study design and detail the relationship between several concepts.

The following theoretical statements were formulated to provide the model with possible solutions to preventing substance use/abuse by student nurses at LCN:Lecturers, professional nurses in clinical facilities, and the college management at various campuses should initiate the prevention of substance use/abuse through counseling, support, and referral to other agencies.Dynamics drive the relationship between student nurses and college management.Psychosocial support increases students’ feelings of self-worth and self-esteem.Various agents, as well as recipients, have an essential role in the prevention of substance use/abuse.The lecturers and professional nurses at clinical facilities are also responsible for creating a healthy relationship with the students and a conducive teaching and learning environment which will not increase students’ stress levels [3,14].There should be ongoing interaction between various health professionals, law enforcement officers, families, and communities to assist each other in preventing substance use/abuse by student nurses at LCN.

### 2.6. Description of the Model 

#### 2.6.1. Socio-Environmental Context

This context includes the community where individual students reside; the clinical environment, which is said to be stressful and demanding; and the legislative framework, such as the education policies, the constitution, and the Liquor Act, all of which contribute to substance abuse by students as the constitution defines 18-year-old individuals as adults. The Liquor Act permits access to alcohol for people who are 18 years and older.

#### 2.6.2. Agents

The agent is the person who facilitates support. In this study, the agents were identified as lecturers, professional nurses, peers, family members, social workers, and psychologists who support students who are abusers or addicts to substances through personal, psychosocial, and therapeutic support. Religious leaders are also essential in counseling and supporting the abuser and family.

#### 2.6.3. Process/Procedure

The procedure is the process of providing students with support from lecturers, professional nurses, and other multidisciplinary stakeholders, enabling students to gain control of their life through student development and capacity-building meetings and workshops, strengthening student policies, and monitoring and evaluating the situation so that students can constructively manage their life and become career-focused and productive in their clinical placement.

#### 2.6.4. Recipients

The recipients are the people who receive support. In this study, the recipients were identified as student nurses who are already affected, and harm reduction was the focus.

Student nurses will be taught how to resist social influences such as smoking cigarettes, drinking alcohol, or using drugs [5]. These peer resistance or refusal skills will be taught within a broader program to enhance general personal and social skills.

#### 2.6.5. Dynamics

The dynamics are the measures taken to assist students who are addicted to substances. The dynamics were identified in this study as counseling, awareness campaigns, and recreation, which could help raise LCN students’ awareness about the dangers of substance abuse and thus prevent substance use/abuse by creating healthy and attractive alternatives that combine and inspire individual life skill development and positive sporting attitudes. Recreation facilities such as playgrounds should be maintained where available and established where possible [25].

#### 2.6.6. Terminus or Outcomes of Prevention

This is portrayed in pink, and there are several factors involved in the end of the prevention process. The model for the prevention of substance use/abuse will yield the following outcomes.

Figure 1: The model structure.

### 2.7. Evaluation of the Model 

Chinn and Kramer’s (2011) steps were utilized for model evaluation. The evaluation of the model, which focused on LCN management, lecturers, and student affairs officers who responded to a self-administered questionnaire, aimed at describing a critical reflection on the developed model. The model was evaluated according to (Chinn & Kramer 2011) the responses of twenty (20) respondents comprising six (6) heads of departments; and eight (8) lecturers, of which one (1) was a senior lecturer with a doctoral degree and three (3) lecturers were busy with their doctoral studies; and two (2) student affairs officers. The model results closely corresponded to real-world values; it was then considered valid. 

Clarity: The meaning and interpretation of words, signs, and sentence structure are clear to comprehend, and similar concepts have been used previously [29]. All evaluators concurred that all concepts were aligned with the principles of clarity. All 20 (100%) respondents agreed that the model clearly describes lecturers’ role in preventing substance use/abuse by the student nurses at LCN. All 20 (100%) respondents agreed that the model needs in-service training on preventing substance use/abuse.

Simplicity: The model’s structure and details are simple to understand, as the signs and symbols are logically connected. All 20 (100%) evaluators agreed that it is simple.

Generality: All evaluators agreed that the model can be used in different settings and sub-populations.

Accessibility: The concepts are easily identifiable, and the purpose is clearly defined, which makes it easier for people to access the model utilizing key concepts. The accessibility of the model was supported. Moreover, all 20 (100%) respondents agreed that the model is accessible because it is free to use.

Importance: The importance of the model is described in [29]. All 20 (100%) respondents agreed that the model addresses the importance of preventing substance use/abuse by the student nurses at LCN. All 20 (100%) respondents agreed that the model is important.

## 3. Conclusions

This study aimed to develop a model for substance use/abuse prevention. The model development was based on the findings from a concept analysis [32], empirical perspective, and conceptual framework [35]. The model can be used to prevent substance use/abuse among students at LCN using a collaborative approach. The steps taken in developing the model were described, and the model evaluation was described using five relevant questions. The researcher recommends that nurse education institutions, clinical preceptors, and health facility managers use and operationalize the model for preventing substance use/abuse in nursing education, nursing research, and nursing practice. The study recommends promoting community education to create public awareness by acknowledging the importance of supporting families early so youth can benefit from growing up in a positive environment.

This will assist in building resistance against delinquent behavior that leads to substance abuse, as stated in [25]. LCN management and the SRC should develop and incorporate policies in the college program. Early intervention by college management should be ensured during orientation or the induction of new students.

Counseling and referral should be provided. The curriculum should be designed to build social and life orientation skills to enhance resistance to substance use/abuse. Recreational facilities should be developed in deprived campuses to keep the students engaged and act as a substitute for substance-related ventures [35]. Participation in sports activities, debates, competitions related to their subjects, dance, and singing to reduce stress and substance use/abuse should be encouraged [36]. Students should also be taught about the practical usage of stipends.

Early identification and intervention of substance abuse/use are critical in any student’s life, though it is not always possible. Approaches that target the students, family, college, and community may be effective preventive measures.

Further studies should be conducted to evaluate the effectiveness of the developed model in preventing substance use/abuse by student nurses and other institutions of higher learning.

## 4. Limitation of the Study

The study was conducted in the Limpopo College of Nursing with a purposive sample of student nurses, lecturers, and college management employees. This limits the transferability of the findings to a broader range of nursing colleges or even other nursing campuses in South Africa.

## Figures and Tables

**Figure 1 healthcare-11-02285-f001:**
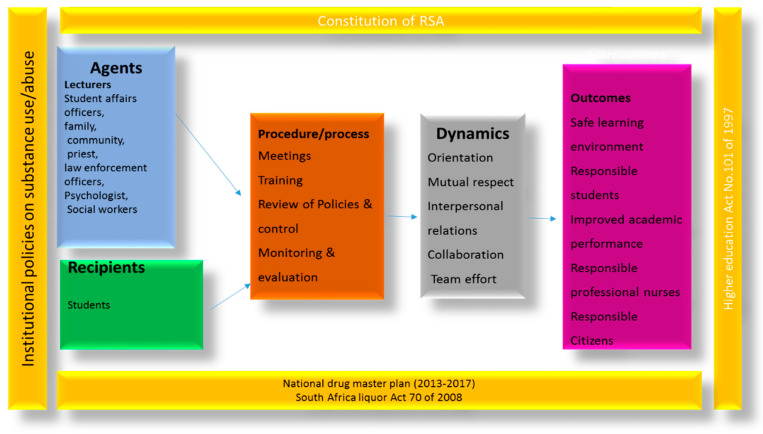
Structure of the model. Source: “A Model to Prevent Substance Use/Abuse by Student Nurses at the Limpopo College of Nursing, Limpopo Province, South Africa.” Ph.D. thesis, unpublished, University of Venda.

**Table 1 healthcare-11-02285-t001:** Results of focus group discussions with students and individual interviews with lecturers based on the prevalence of substances used and problems identified regarding substance use/abuse.

Themes	Identified Central Concept
Contributory factors which lead to substance abuse by student nurses at LCN	Prevention of substance abuse/use
Problems which lead to substance abuse by student nurses at LCN	
Types of substances used and the desired effects	
Challenges induced by substance abuse	
Suggested preventive measures	

Source: Netshiswinzhe, M.D.2018 “Factors contributing to substance use/abuse by student nurses at Limpopo College of Nursing, Limpopo province, South Africa.” Unpublished thesis, 2018, DCur Nursing, University of Venda.

## Data Availability

All data supporting this manuscript have been made available. All data generated or analysed during this study are included in this article.

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
