# Peer review of "A Model to Prevent Substance Use/Abuse by Student Nurses at Limpopo College of Nursing, South Africa"

_healthcare, 2023, doi:10.3390/healthcare11162285_

Round 1

Reviewer 1 Report

The article in question is quite clear and concrete in its objectives. It seems to be a relevant contribution for the adoption of models and practices of substance abuse prevention.

There are, however, some problems that need to be addressed.

More generally, the article lacks contextualization. Virtually nothing is said about the reality and impact of this problem in the South African context, and neither is it explained (nor is empirical evidence presented) as to why the case study presented here (Limpopo College of Nursing). Is it just a representative illustration of a more general pattern? Is it a particular case that deserves, as such, to be further studied? Are there significant differences relative to students from other educational areas? If so, which ones? These choices need to be made explicit and, above all, well grounded.

It should also be noted that the article is very synthetic and schematic, and there is no substantive discussion of results. There is only a description of the construction of a model, and it is not clear how it can be validated without first being subjected to field implementation.

In terms of presenting the study problem, it would be advisable to adopt a less normative tone (examples; destroy the youths, lines 40 and 41; misery, line 47; moral decadence, line 67). Since this is a scientific paper, there should be a more neutral approach in terms of moral position and more focus on analysis.

In more specific terms, the following stands out:

In the case of the abstract, it is not clear what is meant by what is stated in lines 18 to 20 "findings revealed increased prevalence of substance abuse in level one and two students, hence there was a need to develop a model to prevent use/abuse by student nurses at Limpopo College of Nursing" (level one and two??). When, still in the abstract, the results are presented, it does not seem appropriate to assume this content as results. What is written concerns the content of the model at the level of its dimensions/components. About the conclusion, it does not seem appropriate to conclude with what is the intention of the study.

As far as the validation of the model is concerned, it seems debatable to me that one can conclude anything along these lines. For the model to be validated, it needs to be implemented to evaluate its effectiveness. That is not what is presented here. What you call validation is just the stabilization of the model design. When we talk about validation through a self-administered questionnaire, it is said that it includes a critical reflection. Now, what we see is a unanimous assessment by the respondents regarding all the model's evaluation parameters. Besides the fact that it is strange that the agreement is absolute in all parameters, there is nothing to indicate what reflection this model may have provoked.

Finally, and considering the formal aspects, it is not correct to construct some sentences including only the reference numbers. Readability is impaired and there is no reason not to adopt more linguistically correct wording (examples: as attested by 6-10, line 59; development by 30, line 119; Using 26, line 120; proposed by 30, line 132)

Author Response

Dear editors,

Please see the attachment, thank you!

Reviewer 2 Report

The study addresses an important issue, but the results are not clear. Was the protocol applied to a population?

a list of others concerns is below reported:

Abstract: Line 15-16: please, rephrase

Introduction: line 39: the drug consumption among students is officially reported or is an adsumption of the authors? Are there official data explaining the phenomenon?

Line 55-56: Who reported this information? Please, provide the reference.

Line 60: what hubbly- bubbly stands for? Is that a common name of a drug?Please, report the name in brackets and specify what is the substance.

Research problem: Although the report by the close ones is extremely important to suspect the drug abuse, the simple report by voice is not enough strong evidence to consider the subjects as substance abusers. A solid list of evidence should be reported along with data on drug abuse. Furthermore, the authors should give more importance to the severe consequences of drug abuse by a professional health personnel.

Line 87-88: What are the criteria adopted to consider absenteeism and poor performance?

Figure 1: Since the figure is taken from an external source, do the authors have all the permissions? However, the harmonization of fonts.

Method validation: Line 247: please, provide the reference

Although the grammatic is correct, the language should be improved by a specific terms.

Author Response

Dear editor,

Round 2

Reviewer 1 Report

Following revisions by the authors, the article appears suitable for publication.

Author Response

All highlighted similarities corrected

Reviewer 2 Report

the authors sufficiently addressed all the reviewers' concerns and the amendments improved the quality of the paper making it eligible for publication in this journal.

The language level is appropriate 

Author Response

All highlighted similarity corrected